# DiRaGNN: Attention-Enhanced Entity Ranking for Sparse Graph Networks

## Abstract

Sparsity in both the structural and engagement information presents a core challenge in entity ranking problems for graph networks. The interaction dynamics of entities are often characterized by limited structural and engagement information which results in inferior performance of the state-of-the-art approaches. In this work, we present DiRaGNN, an attention-enhanced entity ranking model designed to address the problem of dimension recommendation and ranking for automated watchdogs in the cloud setting. DiRaGNN is inspired by transformer architectures and utilizes a multi-head attention mechanism to focus on heterogeneous neighbors and their attributes. Additionally, our model employs multi-faceted loss functions to optimize for relevant recommendations and reduce popularity bias. To manage computational complexity, we sample a local subgraph that includes multiple hops of neighbors. Empirical evaluations demonstrate significant improvements over existing methods, with our model achieving a 39.7% increase in MRR.

## 1 Introduction

Graph neural networks (GNNs) have proven to be efficient representations for a wide range of real-world systems, including social media graphs (Kipf & Welling, 2016; Hamilton et al., 2017; Borisyuk et al., 2024; Sankar et al., 2021; Zhang et al., 2018; Borisyuk et al., 2024). Heterogeneous graph neural networks (HGNN), on the other hand, offer the additional flexibility to encode both structured and unstructured information associated with various node types, such as explicit links between different nodes and unstructured features associated with nodes, such as texts and images Zhang et al. (2019). HGNNs use either message-passing to learn effective node representations from local graph neighborhoods containing structural relations among nodes and unstructured content Zhang et al. (2019); Hong et al. (2020); Zhao et al. (2021); Hu et al. (2019; 2020), or metapath-based neighbors Wang et al. (2019); Fu et al. (2020); Yun et al. (2019).

**Graph networks for latent entity representation**: Recently, HGNNs have demonstrated promising results in several industrial systems designed for item recommendations in bipartite Ying et al. (2018) or multipartite Yang et al. (2020) user-to-item interaction graphs. GCN Kipf & Welling (2016), GraphSAGE Hamilton et al. (2017), and GAT Velickovic et al. (2017) employ various network architecture and self attention mechanism to aggregate the feature information from neighboring nodes. Further, scalable extensions to these techniques were introduced in Zeng et al. (2019); Chiang et al. (2019); Huang et al. (2018). GRAFRank Sankar et al. (2021) extends GNNs for large-scale user-user social modeling applications and employs multi-modal neighbor aggregators and cross-modality attentions to learn user representations. Yet, entity ranking using heterogeneous graph networks with the structural and engagement is still a core challenge which reduces the quality of the recommended entities. In this work, we investigate this problem in the context of cloud setting and propose an improved framework for enhanced entity ranking. We focus on the message-passing approach as it avoids the need for domain experts to mine meta-paths.

**Entity ranking in cloud setting**: Recommending attributes (dimensions) for aggregating time-series signals to create automated watchdogs that ensure continuous service availability is a complex problem in the cloud setting Surianarayanan & Chelliah (2019); Chen et al. (2020); Montes et al. (2013). Previous research has focused on recommending time-series signals to be associated with automated watchdogs Nair et al. (2015); Srinivas et al. (2024). Generating recommendations in a

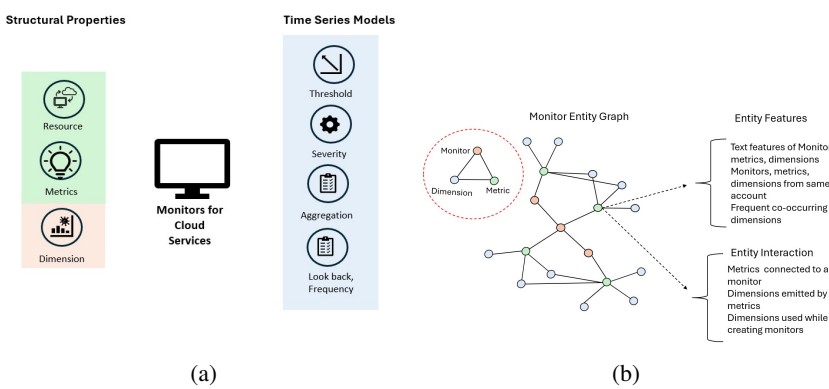

Figure 1: a) Monitors (automated watch dogs) in a cloud setting: Structural elements include resources, metrics, and dimensions which are utilized to raise alerts, b) Monitor entity ranking problem: Nodes represent the monitors, metrics, and dimensions in a cloud setting. Each node contains text features and interacts with their neighboring nodes using link communication.

cloud setting require leveraging domain knowledge and interaction between multiple entities. Therefore, HGNNs are a natural choice due to their effective message passing mechanisms and ability in capturing intricate relationships between entities with different types of relationship. However, applying HGNNs directly to cloud domain datasets is challenging. This is due to the limited structural information available from interactions and the sparse nature of the graph. Most attributes are not connected to all types of nodes, exhibit fewer degrees, and have fewer interactions. Additionally, different neighboring nodes and node attributes contribute differently to learning node presentations, ranking, and predictions.

**Present work**: In this work, we begin by discussing the cloud setting and introducing the different entities involved in the recommendation problem. We then introduce the monitor entity graph and formulate the problem. Next, we discuss the specific nature of the graph, including its long-tailed distribution and the sparsity of interactions. To overcome the challenges of structural and interaction sparsity, we propose leveraging the available set of node attributes and multi-type interactions, and present an attention-enhanced entity ranking framework. Specifically, we make the following contributions.

**Contributions**: We propose, Diverse Ranking for GNN (DiRaGNN), a representation learning framework enhanced with transformer-style graph convolutions. Inspired by transformer architectures, the convolutions incorporate multi-head attention mechanisms, enabling the model to capture complex, long-range dependencies in the graph structure. The edge-aware message passing ensures that the message is sensitive to specific relationship types. Furthermore, the framework incorporates a diversity-aware loss function, which aggregates and attends to various information from different nodes and their interaction patterns. Unlike baseline approaches, DiRaGNN employs a multi-faceted loss function to prioritize relevant recommendations and effectively attend to different node contexts. The diversity loss penalizes similar attention patterns across different heads and learns a comprehensive graph structure, while the ranking loss improves the model's ability to distinguish between more and less relevant entities. Finally, experiments on the monitor-entity dataset show significant improvements in the hit-rate, mean reciprocal rank, and recall over baselines.

## 2 PRELIMINARIES

In this section, we introduce the problem of recommending dimensions for creating monitors for cloud services. We formally define the problem and provide preliminary notations.

### 2.1 PROBLEM FORMULATION

Ensuring continuous availability of services is essential for cloud service providers. Cloud services continuously record information about their health in the form of run-time telemetry. This telemetry

serves as signals to be analyzed for detecting anomalies. Therefore, cloud services are equipped with automated watchdogs, also known as monitors, that monitor service health. Each service emits multiple metrics (time-series data) along different dimensions, which are the topological components of the service. Examples of these dimensions include indicators such as the success of an operation, file path, environment of service deployment, and the identifier of a compute node. Monitors aggregate the metrics emitted along various dimensions, and the alerting conditions operate on the aggregated signal to create an alert. Based on the nature of the monitor, only a subset of the dimensions used for aggregating the signal.

Figure 1a provides an overview of the cloud monitoring framework, including the structural properties of the monitor. These properties encompass the resource upon which the monitor is created (e.g., CPU), the metric being monitored (e.g., processor time), the time series data used to raise alerts, and dimensions such as region and environment of service deployment.

There are several approaches to predict metrics to be monitored for a given cloud service Srinivas et al. (2024). However, given the metric, $m_i$ and the set of dimensions along which the metric is already being emitted, the problem of ranking dimensions along which the metric needs to be aggregated to raise an alert has not yet been explored. Furthermore, the monitor entity graph containing monitors, metrics, and dimensions is a heavy-tailed sparse graph, with limited interaction between most of the monitors, metrics, and dimensions. The sparsity is further exacerbated by the fact that most of the dimensions along which a metric is emitted are not used during monitor creation. As a result, our work focuses on developing a solution to the challenging problem of recommending dimensions for monitor creation in the context of this heavy-tailed sparse graph. We start by modeling the monitor entities and their various interactions, as well as their textual node attributes, as a heterogeneous multi-type interaction graph. From this graph, we generate effective node representations, which are then used for dimension ranking. Next, we define the monitor entity graph and related attributes.

**Definition 1.** *(Monitor Entity Graph): We represent the data as a heterogeneous graph $\mathcal{G} = (\mathcal{V}, \mathcal{E})$ where $\mathcal{V} = \{\mathcal{V}_m \cup \mathcal{V}_d \cup \mathcal{V}_k\}$ represents the set of nodes with $\mathcal{V}_m, \mathcal{V}_d, \mathcal{V}_k$ denoting monitors, dimensions, and metrics, respectively and $\mathcal{E} = \{\mathcal{E}_{md} \cup \mathcal{E}_{kd} \cup \mathcal{E}_{mk}\}$ represents the set of edges, capturing three types of relationships: 1) $\mathcal{E}_{md}$: "monitor associated with dimension", 2) $\mathcal{E}_{kd}$: "metric has dimension", and 3) $\mathcal{E}_{mk}$: "monitor emits metric".*

The graph structure $\mathcal{G}$ captures the intricate relationships in the data that are crucial for making informed recommendations. By explicitly modeling different entity types and their relationships, we enable the model to learn domain-specific patterns. Each node in the graph, denoted as $v \in \mathcal{V}$, has a unique initial representation given by $\boldsymbol{x}_v \in \mathbb{R}^d$, where $d$ is the dimension of the embedding space. The vector $\boldsymbol{x}_v$ is the concatenation of two types of features: intrinsic features, which are domain-specific attributes of the entity (e.g., metric name, dimension name, monitor name, related service), and learned embeddings, which are trainable embeddings that capture the entity's role in the graph structure (e.g., co-occurrence with another node). The monitor entity graph is shown in Figure 1b. It is to be noted that we assume a static graph.

We define the problem of ranking dimensions for monitor creation in the monitor-entity graph, $\mathcal{G}$ with different types of node features and link features as follows:

**Problem** (Dimension Ranking using Heterogeneous Interactions). *Leverage entity features $\{\boldsymbol{x}_v : v \in \mathcal{V}\}$, link features $\{\boldsymbol{e}_{v_1 v_2} \in \mathcal{E}\}$ and monitor entity graph, $\mathcal{G}$, to generate entity representations that facilitate relevant recommendations.*

## 2.2 DEFINITION OF HETEROGENEOUS GRAPH AND NOTATIONS

Graph Neural Networks (GNNs) learn node representations by propagating features from local graph neighborhoods via trainable neighbor aggregators. In this context, we introduce the basic notations and formulations that are useful during message passing in GNN frameworks.

GNNs use multiple layers to learn node representations. At each layer $l > 0$ (where $l = 0$ is the input layer), GNNs compute a representation for node $v_1$ by aggregating features from its neighborhood through a learnable aggregator $F_{\theta,l}$ per layer Hamilton et al. (2017); Kipf & Welling (2016); Velickovic et al. (2017). The embedding for node $v_1$ at the $l$-th layer is given by:

$$\boldsymbol{h}_{v_1,l} = F_{\theta,l}(\boldsymbol{h}_{v_1,l-1}, \{\boldsymbol{h}_{v_2,l-1}\}), v_2 \in N(v_1) \tag{1}$$

The embedding $\boldsymbol{h}_{v_1,l}$ at the $l$-th layer is a non-linear aggregation of its embedding $\boldsymbol{h}_{v_1,l-1}$ from layer $l-1$ and the embedding of its immediate neighbors $v_1 \in N(v)$. The function $F_{\theta,l}$ defines the message-passing function at layer $l$ and can be instantiated using a variety of aggregators Hamilton et al. (2017); Kipf & Welling (2016); Velickovic et al. (2017). The node representation for $v_1$ at the input layer is $\boldsymbol{h}_{v_1,0}$, where $\boldsymbol{h}_{v_1,0} = \boldsymbol{x}_v \in \mathbb{R}^D$.

## 3 ATTENTION ENHANCED ENTITY RANKING FOR SPARSE GNNs

In this section, we begin with the analysis of the monitor entity graph and discuss insights into the structure of the network, as well as the features that impact node relationships. We then use these insights to inform the framework design for dimension recommendations and ranking.

### 3.1 MONITOR ENTITY GRAPH: QUALITATIVE ANALYSIS

Figure 2 illustrates the characteristics of the monitor entity graph. Figure 2a shows the distribution of degree associated with dimensions based on the metric-to-dimension links (i.e., the number of metrics to which each dimension is connected). Similarly, Figure 2b shows the distribution of dimension degree based on the monitor-to-dimension interactions from the monitor entity graph. The likelihood test ratio indicates a resemblance to long-tailed distributions. Next, we analyze the distribution of the percentage of dimensions selected from the set of all dimensions along which the metric is emitted. As seen in Figure 2c, the distribution is skewed to the left, indicating that the majority of monitors do not need to aggregate the metrics along all dimensions along which they are emitted.

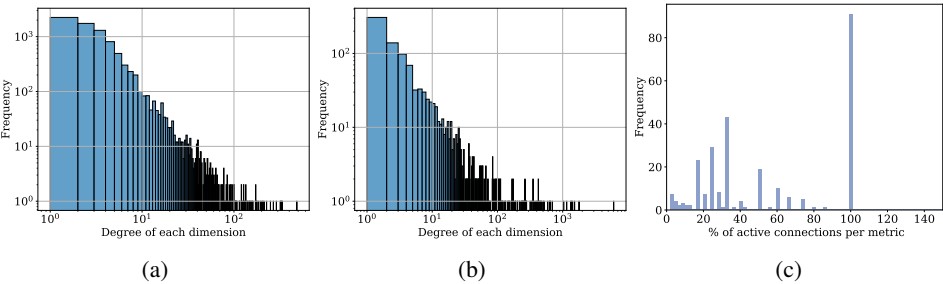

(a)                              (b)                              (c)

Figure 2: Characteristics of the Monitor Entity Graph: a) Distribution of degree associated with dimensions based on the metric-to-dimension links, b) Distribution of dimension degree based on the monitor-to-dimension interactions from the monitor entity graph, and c) Distribution of the percentage of dimensions selected from the set of all dimensions along which the metric is emitted.

**Observation 1.** *The "monitor entity" graph faces activity sparsity. Although many dimensions are associated with metrics, only a subset of them is used to aggregate the metric while raising an alert.*

Next, we analyze the features associated with the nodes to understand its impact on recommendations. We start with the text features associated with the monitor entity graph and its effect on dimensions associated with the monitors. Figure 3a shows the distribution of Jaccard similarity between sets of dimensions associated with monitors that exhibit high cosine similarity ($> 0.8$) between different text features. We consider the text similarity in the monitor names, metric names, and the service account associated with the monitor. The feature embeddings are generated using the "E5" embedding model, a general-purpose model trained through contrastive learning (Wang et al., 2022). The Jaccard similarity of dimensions from monitors exhibits different trends with respect to the similarity in the metric, monitor names, and that from the same service account. Furthermore, the similarity in dimensions with similar monitor names shows higher variance.

In addition, Figure 3b shows the density of pairwise correlation between dimensions connected to a monitor. The correlation plot shows two peaks signifying the presence of distinct groups with

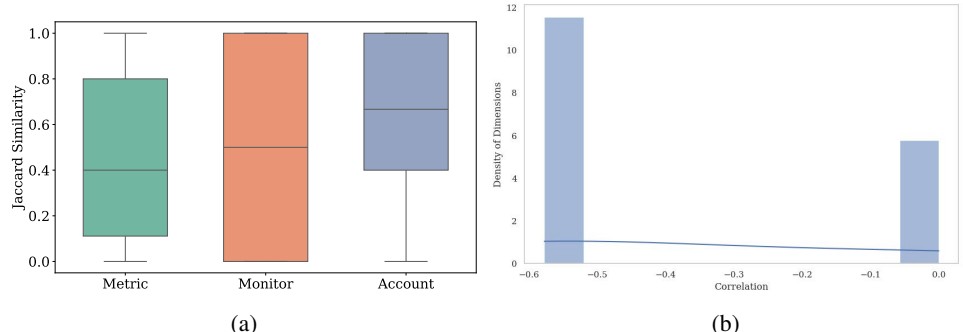

(a)                    (b)

Figure 3: a) Variation in jaccard similarity of set of dimensions associated with monitors with similar metric, monitor names, and same service account, and b) Distribution of pairwise correlation between dimensions.

different trends. As seen in the figure, the majority of dimension pairs are negatively correlated, indicating the absence of a specific dimension in the presence of another.

**Observation 2.** *We observe a significant overlap in the dimensions used by monitors associated with the same service account, those with similar metrics, and those with similar names. However, the extent of similarity varies across the features. Further, the correlation between dimension pairs shows the presence of two distinct groups, where some dimensions are negatively correlated, while the other group is not correlated. Specifically, the framework for representing entities of a node should consider the varying degrees of similarity across different features and the distinct correlation patterns among dimension pairs.*

## 3.2 DiRaGNN for Diverse Ranking of Recommendations

In this section, we present the framework to encode the multifaceted nature of cloud monitoring data and generate node representation for different entities. The graph representation, $\mathcal{G}$ captures both the inherent properties of each entity and its context within the graph structure. The domain-specific attributes provide an inductive bias based on domain knowledge, while the learned embeddings allow the model to discover latent relationships and characteristics. The use of domain-specific attributes enable the model to generalize to new entities not seen during training which is useful in dynamic cloud environments. We begin by discussing the messaging passing mechanism.

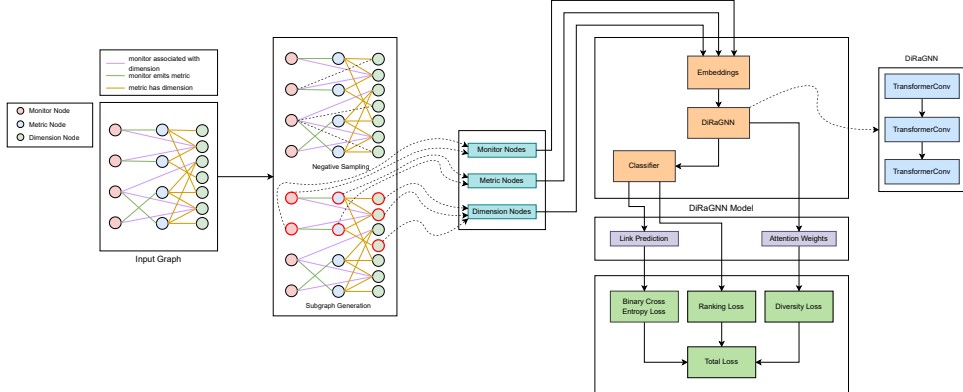

Figure 4: The overall architecture of DiRaGNN framework. The framework uses an enhanced transformer-style graph convolution that incorporates multi-head attention mechanisms, enabling the model to capture complex, long-range dependencies in the graph structure. Additionally, the custom loss function includes diversity loss, ranking loss, and binary cross-entropy loss.

### 3.2.1 Message Passing Mechanism

The framework employs an effective message passing mechanism designed to capture complex relationships and contextual information. The multi-head attention enhanced message passing approach leverages edge-specific transformations and heterogeneous neighborhood aggregation. The key components of our message passing mechanism are:

**Multi-Head Attention.** A multi-head attention mechanism to allow the model to focus on different aspects of node relationships simultaneously. This is important for distinguishing the relevance of different neighbors and relationship types in our graph. The attention weights $\alpha$ are computed as follows: $\alpha_{ij} = \frac{(q_i \cdot k_j)}{\sqrt{d_h d_o}}$, where $q_i$ is the query vector for node $i$, $k_j$ is the key vector for node $j$, $d_h$ is the number of attention heads, and $d_o$ is the output dimension per head. The attention weights are then normalized using softmax: $\alpha_{ij} = \frac{\exp(\alpha_{ij})}{\sum_{k \in \mathcal{N}(i)} \exp(\alpha_{ik})}$ where $\mathcal{N}(i)$ is the set of neighbors of node $i$.

**Edge-Aware Message Transformation.** The message passing considers the type of relationship between the nodes. The transformed message incorporates both the node features and the computed attention weights, ensuring that the message is sensitive to the specific relationship type. The transformed message $m_{ij}$ from node $j$ to node $i$ is computed as $m_{ij} = \alpha_{ij} v_j$, where $v_j$ is the value vector of node $j$. The final aggregated message for node $i$ is $m_i = \sum_{j \in \mathcal{N}(i)} m_{ij}$. The node features are updated as: $x_i^{(l+1)} = \sigma \left( W^{(l)} \cdot \text{CONCAT} \left( x_i^{(l)}, m_i^{(l)} \right) \right)$, where $x_i^{(l)}$ is the feature vector of node $i$ at layer $l$, $W^{(l)}$ is the learnable weight matrix for layer $l$, $\sigma$ is the ReLU activation function and CONCAT is the concatenation operation.

By using attention mechanisms, our approach can dynamically assign importance to different types of relationships and neighbors, important for our problem. The multi-head attention allows the model to leverage limited interactions more effectively, mitigating the impact of sparse data. After multiple rounds of message passing, each monitor node has an enriched representation incorporating information from relevant dimensions and metrics. These representations are used to score potential monitor-dimension pairs.

### 3.2.2 Training Objective

In recommendations systems, the choice of training objective directly influences the model's ability to make accurate, well-ranked, and diverse recommendations. We define a composite loss function that addresses multiple aspects of the recommendation task simultaneously. Our loss function combines three key components: 1) Binary Cross-Entropy Loss, 2) TOP1-max Ranking Loss, and 3) Diversity Loss.

**Binary Cross-Entropy Loss**: The binary cross-entropy (BCE) loss is fundamental for our basic prediction accuracy which is defined as:

$$\mathcal{L}_{\text{BCE}} = -\frac{1}{N} \sum_{i=1}^{N} \left[ y_i \log(\hat{y}_i) + (1 - y_i) \log(1 - \hat{y}_i) \right] \tag{2}$$

where $y_i$ is the true label and $\hat{y}_i$ is the predicted probability. The BCE loss ensures basic prediction accuracy.

**TOP1-max Ranking Loss**: To optimize the order of recommendations, we use the TOP1-max ranking loss proposed in Hidasi & Karatzoglou (2018). This loss aims to push the target score above the scores of negative samples while also acting as a regularizer. It's defined as:

$$\mathcal{L}_{\text{TOP1-max}} = \sum_{j=1}^{N} s_j \left[ \sigma(r_j - r_i) + \sigma(r_j^2) \right] \tag{3}$$

where $r_i$ is the score for the positive sample, $r_j$ are scores for negative samples, $s_j = \text{softmax}(r_j)$, and $\sigma$ is the sigmoid function. The first term of the TOP1-max ranking loss encourages the positive score to be higher than the maximum negative score and the second term acts as a regularizer, pushing negative scores towards zero. This loss optimizes the order of recommendations, crucial for top-k recommendation scenarios.

Table 1: Dataset used in this work

| Dataset | Node | Edge |
|---------|------|------|
| Cloud Monitoring System | # monitors: 18291
# metrics: 4623
# dimensions: 8356 | # monitor, associated_with, dimension: 52148
# metric, has, dimension: 109213
# monitor, emits, metric: 52148 |

**Diversity Loss:** To encourage the model to capture diverse aspects of the graph structure, we introduce a diversity loss component. The diversity loss $\mathcal{L}_{\text{div}}$ is computed as:

$$\mathcal{L}_{\text{div}} = \lambda_{\text{div}} \cdot \frac{1}{L} \sum_{l=1}^{L} \text{MSE}(\alpha_l, \bar{\alpha}_l) \tag{4}$$

where $\alpha_l$ are the attention weights in layer $l$, $\bar{\alpha}_l$ is their mean, and $\lambda_{\text{div}}$ is the diversity strength. This diversity loss penalizes similar attention patterns across different heads, promoting a more comprehensive representation of the graph structure. It prevents the model from over-focusing on a few popular items, particularly important in sparse data settings.

Our final loss function combines all the above components.

$$\mathcal{L}_{\text{total}} = \mathcal{L}_{\text{BCE}} + \mathcal{L}_{\text{div}} + \mathcal{L}_{\text{TOP1}} \tag{5}$$

**Dynamic Loss Balancing** To ensure optimal contribution from each loss component, we use a dynamic loss balancing mechanism which adjusts the weights of different loss components during training, allowing for adaptive optimization of our multi-objective function.

**Neighbourhood Sampling and Subgraph Generation.** In large-scale heterogeneous graphs, processing the entire graph for each recommendation task can be computationally expensive. Moreover, distant nodes in the graph may introduce noise rather than providing useful information. We address these challenges by focusing on the most relevant parts of the graph and employ a multi-stage approach to neighborhood sampling and subgraph generation:

1. We utilize a carefully designed edge splitting strategy for training, validation, and testing, balancing between information propagation and model supervision. During training, negative edges are generated on-the-fly which helps in efficient learning of edge distinction.

2. To capture relevant neighborhood context, we utilize a multi-hop sub-graph sampling method. This approach allows the model to consider both immediate and more distant relationships, crucial for understanding the complex interactions in cloud monitoring systems.

The multi-hop sampling strategy, combined with dynamic negative sampling during training, allows the model to explore a broader range of graph structures while focusing on the most informative negative examples. This also enables the model to effectively learn from sparse interaction graphs, capturing complex relationship between monitors, dimensions, and metrics, while maintaining computational feasibility. Figure 4 summarizes the proposed framework.

## 4 EXPERIMENTS

To evaluate the effectiveness of our proposed framework, we conducted a series of experiments on the task of dimension recommendation for monitors in cloud environments. This section details our experimental setup, evaluation metrics, baseline comparisons, and results analysis.

### 4.1 EXPERIMENT SETUP

#### 4.1.1 DATASETS

Our experiments utilize a heterogeneous graph dataset representing a complex cloud monitoring system. The dataset comprises of three types of entities (nodes) and three types of relationships (edges).

It captures the intricate interactions between monitors, dimensions, and metrics in a cloud environment. The main statistics of the dataset are summarized in Table 1. This graph structure effectively represents the complex relationships in cloud monitoring systems, where monitors are associated with specific dimensions and emit various metrics, while metrics themselves are characterized by multiple dimensions.

**Feature Representation:** To capture the semantic information of monitors and dimensions, we employ feature embeddings generated using a state-of-the-art language model: 1) Monitor Features: Represented by the embeddings of the metric names associated with each monitor. (each monitor is associated with a single metric) and 2) Dimension Features: Represented by the embedding of the dimensions names.

Both feature embeddings are generated using the "E5" embedding model, a general-purpose model trained through contrastive learning (Wang et al., 2022). This approach allows us to capture rich semantic information from the textual descriptions of metrics and dimensions, enabling our model to understand and utilize the contextual relationships between different entities in the cloud monitoring system.

### 4.1.2 TRAINING DETAILS

We train DiRaGNN using $L = 3$ message passing layers with hidden channels size of 256 and output channels size of 128. We use the Adam optimizer with a learning rate of 0.001 and weight decay of $10^{-5}$. To adapt the learning rate during training, we employed a learning rate scheduler which reduces the learning rate by half if the validation loss doesn't improve for 5 consecutive epochs. Our training ran for a maximum of 100 epochs, with early stopping implemented to prevent overfitting. We used a patience of 10 epochs for early stopping. The edge set was divided into training (80%), validation (10%), and test (10%) sets. Within the training set, 70% of edges were used for message passing, and 30% for supervision. For evaluation, we generated fixed negative edges with a ratio of $2 : 1$ (negative to positive). During training, negative edges were dynamically generated using on-the-fly negative sampling. We sampled multiple hops from both ends of a link to create subgraphs. We used a negative sampling ratio of 2.0 during training. We employed a batch size of 128 for training.

### 4.1.3 EVALUATION METRICS

We used the following metrics to evaluate the performance of our model and baselines:

1. **Hit Ratio (HR@k)**: Measures the percentage of test cases where the correct dimension is in the top k recommendations.
2. **Mean Reciprocal Rank (MRR)**: The average of the reciprocal rank of the first correct dimension in the recommendations.
3. **Normalized Discounted Cumulative Gain (NDCG@k)**: Measures the ranking quality of the recommendations, with k set to the number of true dimensions for each monitor.
4. **Recall@k**: The proportion of true dimensions that are successfully retrieved in the top k recommendations.

### 4.2 BASELINES

We compared our model against the following baselines:

- **SAGEConv + Mean**: GraphSAGE Hamilton et al. (2017) with mean aggregation. Element-wise mean pooling for neighbor aggregation and self-embedding concatenation at each layer.
- **SAGEConv + Max**: GraphSAGE with max pooling aggregation. Element-wise max pooling for neighbor aggregation and self-embedding concatenation at each layer.
- **TranformerConv**: The graph transformer operator Shi et al. (2020). A multi-layer Graph Transformer network takes the input to perform attentive information propagation between nodes. For message aggregation, the framework concatenates information across all the heads.

Our proposed model was evaluated in two configurations:

Table 2: Proposed framework outperforms SAGEConv using mean and max aggregation and TransformerConv. The proposed framework demonstrates relative gains of 49.6%, 60.1%, and 29.06% in Hit-Rate@1, NDCG@k, and Recall@5 respectively with respect to the best baseline.

| Metric | HR@1 | HR@3 | HR@5 | MRR | N@k | R@1 | R@3 | R@5 |
|---|---|---|---|---|---|---|---|---|
| SAGEConv + Mean | 0.383 | 0.186 | 0.127 | 0.499 | 0.328 | 0.218 | 0.379 | 0.474 |
| SAGEConv + Max | 0.291 | 0.154 | 0.111 | 0.414 | 0.262 | 0.165 | 0.323 | 0.398 |
| TransformerConv | 0.331 | 0.188 | 0.134 | 0.481 | 0.306 | 0.178 | 0.399 | 0.523 |
| TransformerConv + Diversity Loss | 0.547 | 0.238 | 0.157 | 0.650 | 0.500 | 0.21 | 0.561 | 0.655 |
| **DiRaGNN** | **0.573** | **0.246** | **0.159** | **0.672** | **0.525** | **0.342** | **0.592** | **0.675** |

- **Diversity Loss**: Our model featuring a custom TransformerConv with multi-head attention and integrated layer-wise diversity loss.

- **DiRaGNN: Diverity Loss + Ranking Loss**: Our complete model featuring a custom TransformerConv with integrated layer-wise diversity loss and ranking optimization.

## 4.3 RESULTS

Both variants of our proposed model significantly outperform the baselines across all metrics. The full model with both diversity and ranking losses shows the best overall performance. Our full model achieves HR@1 of 0.573, which is a 49.6% improvement over the best baseline (SAGEConv + Mean at 0.383), indicates a substantial enhancement in the ability to recommend the most relevant dimension as the top choice. The NDCG@k score of 0.525 for our full model, compared to 0.328 for the best baseline, represents a 60.1% improvement. The results suggests that our model not only recommends relevant dimensions but also ranks them more effectively. Our model shows significant improvements in Recall@k, particularly at higher k values. The Recall@5 of 0.675 for our full model compared to 0.523 for TransformerConv indicates a 29.06% improvement in retrieving relevant dimensions within the top 5 recommendations. The introduction of diversity loss alone leads to substantial improvements across all metrics compared to the baselines. This underscores the importance of encouraging diverse attention patterns in the model. The addition of ranking loss to diversity loss results in further improvements, particularly in HR@1 and MRR. This highlights the effectiveness of our multi-faceted loss function in optimizing both accuracy and ranking quality.

Next, we discuss the intuition behind the improved performance of the proposed framework;

1. DiRaGNN uses a multi-head attention mechanism, which allows it to capture the relative importance of different neighbor types more effectively than "SAGEConv", which relies on mean or max aggregation. On the other hand, while "TransformerConv" uses attention, it may struggle to distinguish between different types of relationships as effectively as DiRaGNN.

2. The diversity loss in framework encourages it to capture varied aspects of the data, whereas the baselines do not have an explicit mechanism to encourage diversity in their representations.

3. The ranking loss from DiRaGNN directly optimizes for ranking quality. In contrast, the baselines are typically trained with binary classification objectives, which may not directly optimize for ranking quality.

4. Baselines may struggle with sparsity more than the proposed framework, as they rely more heavily on dense connection patterns.

## 4.4 ABLATION STUDY

To understand the individual contributions of our model's key components, we conducted an ablation study focusing on the diversity loss and ranking loss. We compared three model variants: 1) Base model (without diversity and ranking loss), 2) Model with only ranking loss, and 3) Model with

only diversity loss. We aggregate the changes in the ranks of the recommended dimensions across different monitors and the performance of these variants using rank stability plots, which visualize change in the relevance of top ranked dimensions across different model configurations.

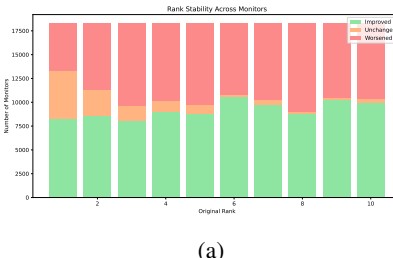 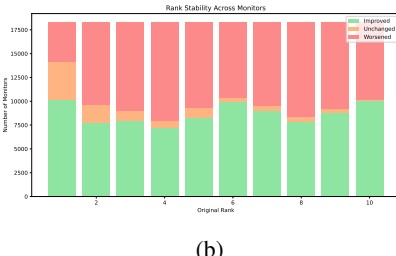

(a)                                    (b)

Figure 5: Impact of Ranking and Diversity Loss on Model Performance: a) Impact of Ranking Loss on Model Performance, and b) Impact of Diversity Loss on Model Performance. Overall quality of recommended dimensions improved with the addition of the ranking loss and diversity loss.

**Impact of Ranking Loss.** Figure 5a illustrates the effect of incorporating only the ranking loss into our base model. The results show a significant improvement in ranking performance across all positions: 1) Top-ranked items (Rank 1) demonstrate high stability, with a large proportion remaining unchanged, 2) Lower-ranked items (Ranks 2-10) show substantial improvements, with the proportion of improved rankings increasing for initially lower-ranked items, and 3) Very few rankings worsen, especially in the lower ranks, indicating that the ranking loss effectively optimizes the order of recommendations. The ranking loss appears to be particularly effective at improving the position of dimensions initially ranked lower (Ranks 8-10), suggesting that it helps in surfacing relevant but previously undervalued dimensions.

**Impact of Diversity Loss.** Figure 5b illustrates the effect of incorporating only the diversity loss into our base model. While also improving overall performance, it exhibits a different pattern compared to the ranking loss: 1) As compared to the impact of the ranking loss, improvements are more uniformly distributed, indicating a more uniform impact on the entire ranking, and 2) There is slightly less stability for top-ranked items compared to the ranking loss model.

The diversity loss appears to encourage a broader exploration of the dimensional space, promoting a wider range of dimensions across different rank positions. Both losses contribute positively to the model's performance, but with distinct characteristics. The ranking loss is crucial for optimizing the order of recommendations, particularly beneficial for surfacing relevant dimensions from lower ranks. The diversity loss contributes to a more balanced improvement across all ranks, likely enhancing the model's ability to recommend a varied set of relevant items. Both losses combined provide a synergistic effect: the ranking loss optimizes the overall order, while the diversity loss ensures a broader range of relevant dimensions to be considered.

These findings support our decision to incorporate both ranking and diversity losses in the final model. The ranking loss ensures high-quality ordered recommendations, while the diversity loss helps prevent over-focusing on a narrow set of popular items - a crucial factor in sparse interaction graphs typical in cloud monitoring systems.

## 5 CONCLUSION

We propose a novel recommendation framework, DiRaGNN, which leverages the available set of node attributes and multi-type interactions to overcome the challenges of structural and interaction sparsity in the monitor entity graph. DiRaGNN incorporates a diversity-aware loss function, edge-aware message passing, and multi-head attention mechanisms inspired by transformer architectures. Experiments on the monitor-entity dataset demonstrate significant improvements in hit-rate, mean reciprocal rank, and recall over baseline approaches. The proposed framework presents a promising approach for addressing the recommendation problem in cloud settings with sparse and diverse interaction data.

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
