# OpenReview forum: "DiRaGNN: Attention-Enhanced Entity Ranking for Sparse Graph Networks"
_ICLR.cc/2025/Conference — ICLR 2025 Conference Withdrawn Submission_

### Official Review · Reviewer_LpRQ · 2024-10-30

**Soundness:** 1
**Presentation:** 1
**Contribution:** 1
**Rating:** 1
**Confidence:** 4

**Summary:**

Cloud service providers (e.g., Azure, AWS, GCP) aim to ensure the continuous availability of their cloud services. The cloud service monitors, also known as watchdogs, continuously monitor the status of cloud services, tracking various metrics and logs to detect anomalies. In this work, the authors represent a cloud service network as a heterogeneous monitor entity graph. In the context of this paper, the heterogeneous graph consists of three types of nodes: monitor, metric (such as device usage or latency), and dimension (like an attribute of a device). It also contains three types of edges representing the connections between these node types. The paper utilizes the heterogeneous graph to derive the embedding vectors for each node in the monitor entity graph. These embedding vectors are then used to compute a composite loss function, which combines BCE Loss, Ranking Loss, and Diversity Loss. (However, the paper does not provide detailed information on the inputs for these losses.)

**Strengths:**

- The authors proposed the DiRaGNN framework for the dimension recommendation and ranking problem in the context of cloud services.
- The authors address the computational challenges of processing large-scale heterogeneous graphs by employing neighborhood sampling and subgraph generation.
- The author evaluates the diversity loss and ranking loss through an ablation analysis, comparing three model variants and visualizing how the relevance of top-ranked dimensions changes across configurations using rank stability plots. However, providing additional quantitative metrics for the rank stability plots would further help reviewers understand the differences between the model variants.

**Weaknesses:**

- The paper writing is poor. Almost every part of the paper lacks critical information that would help the reviewer understand the work, making the entire paper filled with ambiguity. Moreover, the authors did not clearly introduce or define what the “dimension ranking problem” (or the “entity ranking problem in a cloud setting”) is. It was only after reading a cited paper, Intelligent Monitoring Framework for Cloud Services: A Data-Driven Approach, that I could finally understand what the authors meant by the “dimension recommendation and ranking problem.”
- The font size in the figures is too small (Figure 1, Figure 3b, Figure 4, and Figure 5), making it very difficult to read. Additionally, the figures are poorly designed, looking almost like children’s doodles. (For example, in Figure 1a, the elements in the blocks are not properly aligned, and in Figure 1b, the word “dimension” is even tilted rather than being properly displayed in the horizontal direction.)
- While the paper introduces the adopted DiRaGNN framework and loss functions, the authors fail to explain the source of the datasets used. Furthermore, the ground truth for the dimension ranking task for monitors is neither defined nor explained. Without these details, it becomes difficult for the reviewer to assess the capability of the proposed DiRaGNN framework, relying solely on the reported metric scores.
- The content of subsubsection 3.2.1 (MESSAGE PASSING MECHANISM) actually describes the concept of a Heterogeneous Graph Attention Network (referenced below). However, the authors do not cite any relevant papers to support this approach.
- In the framework overview shown in Figure 4, the DiRaGNN framework includes both a classifier and link prediction. However, the paper does not explain how the loss is calculated through these two components, making it difficult to understand the framework's workflow.
-The authors only conduct experiments on a single dataset. Performing experiments on additional datasets would help demonstrate the generalizability of the proposed approach.
- The paper’s presentation could be improved; figures are not vector graphics, and some tables appear poorly formatted.

**Questions:**

- There is a family of graph neural networks called heterogeneous graph neural networks. In addition to the well-known Heterogeneous GAT (HGAT), other models include HGT and HetGNN (referenced below). These methods can learn node representations for heterogeneous graphs and inherently handle multiple edge types. Have you ever tried using these methods as baseline models to compare with DiRaGNN?

---

### Official Review · Reviewer_S1qQ · 2024-11-02

**Soundness:** 2
**Presentation:** 2
**Contribution:** 1
**Rating:** 3
**Confidence:** 4

**Summary:**

In this paper, the authors propose an attention-enhanced entity ranking model aimed at improving dimension recommendation and ranking systems. However, the technique employed is not particularly innovative, as there are several existing studies incorporating transformer architectures into heterogeneous graphs. Furthermore, the authors do not provide strong experimental results in comparison to recently proposed methods.

**Strengths:**

1. This paper introduces a diverse ranking approach for GNNs by incorporating multi-head attention to capture long-range dependencies within the graph structure.
2. Overall, the paper is well-written.

**Weaknesses:**

1. The paper’s key novelty is unclear, given that previous work has already integrated transformer architectures into GNNs.
2. The experimental results are limited, as the authors only include SAGEConv and TransformerConv as baselines and use a single heterogeneous graph dataset (without a publicly accessible link).
3. The intuition behind the proposed methods, particularly concerning the monitor entity graph, is not clearly explained.

**Questions:**

My main concerns are: (i) the distinctions between the proposed method and existing attention-based graph methods, such as [1][2][3]; (ii) the underlying intuition for the proposed approach, especially the specific design choices related to the monitor entity graphs; and (iii) the limited experimental scope, which includes only one private dataset and two baseline methods, making it difficult to assess the proposed method’s advantages.

In particular, incorporating attention mechanisms into graphs has been extensively studied in prior work [1][2][3]. I strongly recommend that the authors provide a detailed discussion of how their approach differs from previous methods. Additionally, it is crucial to evaluate the method across various graph datasets, as real-world graphs often exhibit diverse structures that can better test the generalization capabilities of the proposed model. Thus, I suggest that the authors compare their method on some publicly available graph datasets.

[1] Heterogeneous Graph Attention Network
[2] Graph Transformer Networks
[3] NodeFormer: A Scalable Graph Structure Learning Transformer for Node Classification

---

### Official Review · Reviewer_oxVf · 2024-11-03

**Soundness:** 2
**Presentation:** 2
**Contribution:** 1
**Rating:** 5
**Confidence:** 4

**Summary:**

This paper focuses on the problem of recommending dimensions for monitor creation in the cloud setting, where an attention-enhanced entity ranking model is proposed. The authors illustrate the characteristics of the monitor entity graph, then study a set of loss functions to improve the recommendation quality, and finally empirical results show the improvements over classic baselines (e.g., SAGEConv).

**Strengths:**

This paper is easy to read and well organized.

In section 2 and 3, the authors spend 2.5 pages to formulate the research problem and illustrate the characteristics of the monitor entity graph, which might be helpful for the beginners to interpret the topics of this work.

**Weaknesses:**

1. The section of related works is missing. I highly recommend the authors to compare this work with recent advances in the filed of representation learning over heterogeneous graphs and emphasize the technique contributions.

2. It seems that DIAGNN is a combination of existing works, and thus the contribution looks incremental. Meanwhile, it is not clear to me which part of the networks is proposed to solve the issue of graph sparsity and why it works.

3. The training objective, including CE loss, TOP1-max Ranking loss and diversity loss, is not new. Also, the idea of subgraph sampling with negative training examples is standard in task of node classification and recommender systems. The authors are encouraged to clarify the difference to existing works.

**Questions:**

Please check the comments above.

---

### Official Review · Reviewer_8mcU · 2024-11-04

**Soundness:** 2
**Presentation:** 3
**Contribution:** 2
**Rating:** 5
**Confidence:** 3

**Summary:**

The paper presents "DiRaGNN," an attention-enhanced entity ranking model designed to address the challenges of sparsity in structural and engagement information for entity ranking problems within graph networks. DiRaGNN leverages a transformer-inspired multi-head attention mechanism to focus on heterogeneous neighbors and their attributes. The model employs a multi-faceted loss function that includes diversity-aware and ranking losses to improve recommendation relevance and reduce popularity bias. Experimental results show that DiRaGNN significantly improves entity ranking accuracy, achieving a 39.7% increase in mean reciprocal rank (MRR) over existing approaches.

**Strengths:**

1. The proposed model is explicitly designed to tackle the issue of sparse interactions, which is a common limitation in entity ranking tasks for practical consideration.
2. The paper is generally written with good clarity and thus is easy to follow.

**Weaknesses:**

1. The technical contribution of this work appears limited, particularly in the design of the DiRaGNN model, which could be seen as fitting within the general framework of graph transformers. The authors are encouraged to emphasize their unique technical innovations to better distinguish their approach.
2. Some illustrations, such as Figure 4, are difficult to interpret. The authors are advised to improve the visual clarity of these figures to enhance readability.
3. The process of neighborhood sampling and subgraph generation lacks sufficient detail. For instance, the "carefully designed edge splitting strategy" should be elaborated upon. Additionally, for the multi-hop subgraph sampling method, it is unclear whether it was achieved through random walks or simple node sampling.
4. In terms of evaluation, only one medium-sized dataset was used, which may limit the generalizability of the results. Moreover, the paper does not specify the number of repeated experiments or the number of seeds used, which is crucial for demonstrating the robustness of the performance.

**Questions:**

Please see the above weaknesses for details.

---

### Note · Authors · 2024-11-22

I have read and agree with the venue's withdrawal policy on behalf of myself and my co-authors.